# Benchmarking Data-driven Surrogate Simulators for Artificial Electromagnetic Materials

**Yang Deng**[*]
yang.deng@duke.edu

**Juncheng Dong**[*]
juncheng.dong@duke.edu

**Simiao Ren** [*]
simiao.ren@duke.edu

**Omar Khatib**
omar.khatib@duke.edu

**Mohammadreza Soltani**
mohammadreza.soltani@duke.edu

**Vahid Tarokh**
vahid.tarokh@duke.edu

**Willie J. Padilla**
willie.padilla@duke.edu

**Jordan M. Malof**
jordan.malof@duke.edu

Department of Electrical and Computer Engineering
Duke University
Durham, NC 27708

## Abstract

Artificial electromagnetic materials (AEMs), including metamaterials, derive their electromagnetic properties from geometry rather than chemistry. With the appropriate geometric design, AEMs have achieved exotic properties not realizable with conventional materials (e.g., cloaking or negative refractive index). However, understanding the relationship between the AEM structure and its properties is often poorly understood. While computational electromagnetic simulation (CEMS) may help design new AEMs, its use is limited due to its long computational time. Recently, it has been shown that deep learning can be an alternative solution to infer the relationship between an AEM geometry and its properties using a (relatively) small pool of CEMS data. However, the limited publicly released datasets and models and no widely-used benchmark for comparison have made using deep learning approaches even more difficult. Furthermore, configuring CEMS for a specific problem requires substantial expertise and time, making reproducibility challenging. Here, we develop a collection of three classes of AEM problems: metamaterials, nanophotonics, and color filter designs. We also publicly release software, allowing other researchers to conduct additional simulations for each system easily. Finally, we conduct experiments on our benchmark datasets with three recent neural network architectures: the multilayer perceptron (MLP), MLP-mixer, and transformer. We identify the methods and models that generalize best over the three problems to establish the best practice and baseline results upon which future research can build.

## 1 Introduction

Artificial electromagnetic materials (AEMs) are a class of materials that derive their electromagnetic properties primarily from their physical structure rather than their chemistry. With the appropriate structural design, AEMs have achieved exotic properties that are not realizable with conventional

---

[*]Authors contributed equally

35th Conference on Neural Information Processing Systems (NeurIPS 2021) Track on Datasets and Benchmarks.

materials, including invisibility cloaking [1] and negative refractive index [2]. AEM research comprises a large community, now encompassing major areas of research around specific classes of AEMs such as metamaterials[2], photonics[3] and plasmonics[4].

One invaluable tool in AEM research is the ability to evaluate their electromagnetic properties, $s$, based upon that AEM's structure, $g$. This capability is crucial to enable scientific exploration, i.e., AEM design and many other activities. Although Maxwell's equations fundamentally govern the properties of AEMs, for advanced designs, there is often no explicit mathematical relationship between $s$ and $g$, or it is not yet known. In these scenarios, researchers can still utilize computational electromagnetic simulation (CEMS) to obtain estimates of the AEM properties for a *specific* design. CEMS is a form of scientific computing that relies upon numerically solving Maxwell's equations to obtain the (complex-valued) electric and magnetic fields over a 3-dimensional mesh. This process is computationally intensive, however, and it must be repeated from scratch for each variation of AEM structure under consideration - sometimes millions or (recently) billions [5] - creating a major research bottleneck.

In recent years, it has been shown that machine learning models - especially deep neural networks (DNNs) - can leverage (relatively) small datasets of design-property pairs from CEMS simulation, $D = \{g_i, s_i\}_{i=1}^N$, to infer computationally efficient surrogate models of the CEMS process (e.g., $s = f(g)$). Although these models require an initial investment of simulations, once trained, they are often several orders of magnitude faster than CEMS (e.g., $10^5$ times faster [5]). This substantially mitigates the computational bottleneck imposed by CEMS, accelerating research for many AEM systems and bringing problems of unprecedented complexity within reach.

The use of data-driven surrogate models has grown rapidly in AEM research over the past several years [6]. Fig. 1 illustrates this growth for DNN-based publications - the dominant model, by far. Despite this success, however, there is a near-complete absence of replication and benchmarking of models across different studies, making it difficult to determine whether, or to what extent, real scientific progress is being made over time and invested research funds. Among the studies compiled in a recent review of this literature [6], we find that no dataset has yet been utilized in more than one study (i.e., no replication), and the specific architectures, validation schemes, and performance metrics employed for surrogate models varied widely.

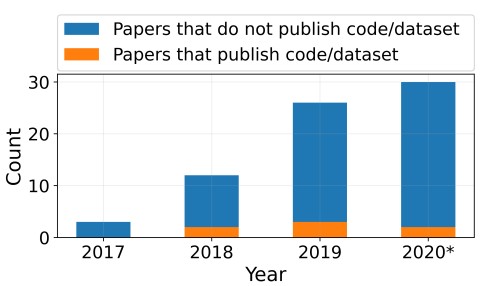

Figure 1: The number of papers published that use a data-driven surrogate method in the AEM community, cited in a recent review [6]. *Note that in 2020 the count is incomplete, as the review ended its literature review section in 2020.

One fundamental reason for the lack of replication and benchmarking in AEM research may be the difficulty of reproducing the CEMSs from a previous study, which requires substantial expertise and computation time. Furthermore, the details needed to reproduce the CEMS are incomplete. One potential solution to this problem is for authors to release their CEMS datasets and models. However, as illustrated in Fig. 1, thus far, this has rarely been done in AEM research. The absence of replication and benchmarking has been noted in recent reviews on AEM research [6, 7], however, to our knowledge no studies have yet to address this problem.

## 1.1 Contributions of this Work

In this work we address these problems by developing a benchmark dataset of three diverse AEM simulation problems: metamaterials, nanophotonics, and color filter designs. These problems were adopted from recently-published AEM research, helping to ensure their significance and relevance to the broader AEM community. This also enabled us to include models employed by the original authors of each dataset in our benchmark experiments, strengthening our results and establishing a precedent of comparison with prior work. We also included problems from several unique sub-fields within AEM research, with the goal of again increasing relevance, and increasing likelihood that any conclusions drawn from our benchmark will generalize across AEM problems.

We then employed our benchmark to study three state-of-the-art DNN architectures: the multilayer perceptron (MLP), MLP-Mixer [8], and Transformer [9]. We systematically optimize and compare these three models to determine whether a particular architecture generalizes best for solving AEM problems (i.e., Maxwell's equation). Through our optimization process, we also compare a variety of hyperparameter and processing choices that have been considered in the AEM literature (e.g., batch normalization [10] and skip connections [11]), but until now had not been systematically compared. We summarize the contribution of this work as follows:

1. *We create the first publicly available and easily accessible set of benchmark surrogate modeling problems for the AEM community.* (i) This benchmark will enable researchers to identify the best models and practices, and act as means to measure research progress in the field. (ii) By establishing this benchmark we hope to encourage greater replication and comparison in the AEM community, as well as practices of sharing code and simulation data.

2. *We conduct the first benchmarking of data-driven surrogate models for the AEM community.* Using our new benchmark we systematically compare several state-of-the-art models, providing initial guidance to the AEM community on the types of models and training practices that work best to solve scientific computing problems involving Maxwell's equation. To our knowledge this is the first replication, or comparison against, prior data-driven surrogate modeling results within the AEM community. Our benchmarking is also the first experimentation with Transformers or MLP-Mixers to solve AEM surrogate modeling problems as well.

## 2 Related Work

The AEM simulation problem considered here is a special case of scientific computing, which has become a ubiquitous tool in science [12]. In particular, data-driven surrogate models (e.g., DNNs) have been proven effective as solutions to a variety of scientific computing problems, including the approximation of solutions to partial differential equations [13, 14, 15, 16, 17].

AEM research is primarily concerned with approximating solutions to Maxwell's equations, parameterized by the initial conditions of the problem (i.e., the structure of the AEM). Nearly all recent data-driven surrogates in the AEM research employed conventional feed-forward DNNs, sometimes also called multi-layer perceptrons (MLPs). The main differences between these models are variations on common practices, such as the addition or exclusion of skip connections [18], batch-normalization [19], dropout [20], etc [21, 22, 23, 24, 25, 26, 27]. More details of the current state of machine learning enabled AEM simulation can be found in recent reviews [28, 29, 7, 30]. Therefore, in this work we adopt the MLP as a representative baseline model for our benchmarking experiments, and we systematically tested all those variations in our optimization.

As discussed in Section 1.1, despite a large number of publications on this topic, few researchers have published their software, models, or datasets [31]; and we have not found any public benchmark problems. These limitations of existing AEM research have been noted in recent publications, including review articles [6, 7]. Therefore, we aim to provide the first public benchmark in this community, with an accessible suite of software and data, as well as standardized evaluation metrics.

## 3 Background and Problem Formulation

### 3.1 AEM properties and Maxwell's Equations

The electromagnetic properties of AEMs are governed by Maxwell's equations, a set of partial differential equations (PDEs), given as,

$$\nabla \times \mathbf{E} = -\frac{\partial \mathbf{B}}{\partial t}, \qquad \nabla \times \mathbf{H} = \frac{\partial \mathbf{D}}{\partial t} + \mathbf{J}, \qquad \nabla \cdot \mathbf{D} = \rho, \qquad \nabla \cdot \mathbf{B} = 0 \qquad (1)$$

where $\mathbf{E}$ is the electric field strength (V/m), $\mathbf{B}$ is the magnetic flux density (V·s/m$^2$), $\mathbf{J}$ is the electric current density (A/m$^2$), and $\rho$ is the electric charge density (A·s/m$^3$). $\mathbf{D}$ and $\mathbf{H}$ are the auxiliary fields, and sometimes referred to as the electric displacement (A·s/m$^2$), and the magnetic field strength

(A/m), respectively. The bold characters denote time-varying vector fields and are real functions of spatial coordinates $\mathbf{r}$, and time $t$. The relationship between $\mathbf{E}$, $\mathbf{B}$ and $\mathbf{D}$, $\mathbf{H}$ is given by the constitutive equations, and the general case (ignoring magneto-optical coupling) [32] can be written as:

$$\mathbf{D}(\omega) = \bar{\bar{\varepsilon}}(\omega) \cdot \mathbf{E}(\omega), \qquad \mathbf{B}(\omega) = \bar{\bar{\mu}}(\omega) \cdot \mathbf{H}(\omega) \tag{2}$$

where $\omega$ is the angular frequency, and $\bar{\bar{\varepsilon}}(\omega)$ and $\bar{\bar{\mu}}(\omega)$ are rank 2 tensors termed the permittivity and permeability, respectively. In nature, all materials have their own distinctive – and relatively simple – $\varepsilon$ and $\mu$ values (dropping the explicit frequency dependence and tensor notation) that directly determine the three-dimensional field solutions to Maxwell's equations. When solving Maxwell's equations in a 3D space, we need to know two main elements: the incident field and the material properties $(\varepsilon, \mu)$ [33], which is given by the AEM in our case. With the set up of accurate boundary conditions in the material space, it has been shown that we can find analytical solutions in a 3D space for simple geometries [34, 35]. In AEM design, the permittivity and permeability are determined by the AEM geometry, instead of the chemistry or band-structure of the constituent elements, thus resulting in effective optical constants – denoted by $\varepsilon_{eff}$ and $\mu_{eff}$. Referring to Eqs. 1 and 2, the prescription for AEM design using CEMS can be written as:

$$m(\mathbf{r}), \varepsilon(\mathbf{r}, \omega), \mu(\mathbf{r}, \omega) \quad \rightarrow \quad \mathbf{E}(\mathbf{r}, \omega), \mathbf{H}(\mathbf{r}, \omega) \quad \rightarrow \quad s(\omega) \tag{3}$$

where $s(\omega)$ denotes the resulting electromagnetic scattering, which may be the reflectance $R(\omega)$, the transmittance $T(\omega)$, or the absorptance $A(\omega) = 1 - T(\omega) - R(\omega)$, etc. The size of the geometry mesh $m(\mathbf{r})$ and material properties $\varepsilon(\mathbf{r}, \omega), \mu(\mathbf{r})$ is determined by the mesh of the AEM, which is of order $\mathbf{r} \sim 10^6$. Although the resulting fields $\mathbf{E}(\mathbf{r}, \omega), \mathbf{H}(\mathbf{r}, \omega)$ are also of order $\mathbf{r}$, the scattering $s(\omega)$ is typically a size of $\omega \sim 10^3$.

## 3.2 Computational Methods to Solve Maxwell's Equations

It becomes increasingly difficult to map the geometry $g$ to its property $s$ analytically as the geometry grows more complex. Conventionally, we either fabricate the AEM and experimentally measure its scattering $s$ or use CEMS to determine the geometry required to give a desired $s$. The CEMS is a much more preferred approach than experimental verification because of the much lower total time and cost. Most CEMS utilize Finite Element Method (FEM) [36], Finite Difference Time Domain (FDTD) [37], or Finite Difference Frequency Domain (FDFD) [38] methods to solve partial differential equations within the defined boundary conditions to find solutions to Maxwell's equations.

## 3.3 Data-driven Surrogate Models

Although CEMS can accurately solve Maxwell's equations to provide electromagnetic properties $s$ for AEMs, their computational time can rise exponentially as AEMs turn into free-form geometry. On the other hand, the data-driven surrogate models can obtain electromagnetic properties $s$ of interest in milliseconds. In training the data-driven surrogate models, we simplify the steps down to mapping a parameterized geometry $g$ of AEMs as input to desired property and feed the surrogate model geometry parameters as input $g$ and desired property as target $s$. The simplification of the problem for the surrogate model also avoids the complexity of handling 3D inputs of geometry or fields as 3D vectors. The data-driven approach can be written as::

$$g \quad \rightarrow \quad \text{Surrogate Model} \quad \rightarrow \quad s(\omega) \tag{4}$$

where the surrogate model is $\hat{f}$ learned by the neural network through training on geometry spectral pairs $D = \{g_i, s_i\}_{i=1}^N$, i.e. $s = \hat{f}(g)$.

# 4 Benchmark Design and Resources

The objective of our benchmark is to establish a shared set of problems on which the AEM research community can compare data-driven surrogate models and thereby demonstrate and measure research progress. To achieve this goal, we chose three initial problems to include in our benchmark (to be expanded over time), and we share resources to make adoption and replication of these benchmarks easy for the AEM research community.

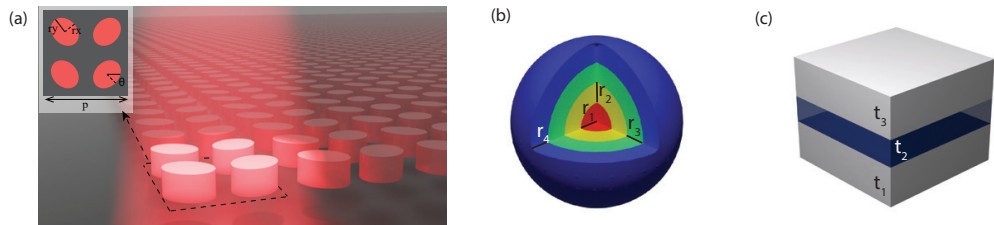

Figure 2: Schematics of geometry in three physical problems. (a) Infinite array of all-dielectric metasurfaces consists of four elliptical-resonators supercells. (b) A nanophotonic particle consists of four layers. (c) The color filter design with one layer of $SiO_2$ sandwiched between two metal layers of $Ag$. From bottom to top, each layer has thickness $t_1 - t_3$.

## 4.1 AEM Benchmark Problems and Selection Criteria

The three AEM problems that we selected for inclusion in our benchmark are presented in Table 1, along with key details. These problems were chosen based upon several criteria to maximize relevance to, and adoption by, the AEM research community. The first criterion was that the dataset was introduced in a recent AEM publication. This helps ensure that these are significant problems and problems that are of interest to the broader AEM community. This also establishes a precedent of replicating and comparing existing work rather than introducing more novel problems. For example, this enabled us to compare our models to those employed by the original authors of each dataset, strengthening our benchmarking results (see Section 5).

A second criterion was representativeness; we deliberately chose problems from different sub-fields within AEM research. This was done to broaden the relevance of the problems to the AEM community as a whole and ensure that any models are evaluated on problems that span a variety of physical systems, making any conclusions more likely to generalize. Finally, we chose a set of problems that would also span the range of complexity currently observed in the AEM research. More precisely, we chose problems of varying input and output dimensionality, since dimensionality is an influential factor in the performance and behavior of machine-learning models. Below we describe major details about each benchmark problem, however, further details can be found in Section 1 of the supplement.

Table 1: Overview of the artificial electromagnetic material datasets. $D_{in}$ and $D_{out}$ represent the dimensionality of the input and output of each dataset, respectively. CPU time is estimated from total time it takes to run on a 16 core AMD CPU machine.

| Dataset | $D_{in}$ | $D_{out}$ | Sub-area | Simulations | CPU time |
|---|---|---|---|---|---|
| All-dielectric metasurface | 14 | 2001 | Metamaterials | 60,000 | 7 months |
| Nanophotonic particle | 8 | 201 | Nanophotonics | 50,000 | 1.5 hours |
| Color filter | 3 | 3 | Optical waveguides | 100,000 | - |

**All-dielectric metasurfaces. (ADM)** This problem was originally described and published in [39]. The ADM benchmark dataset [39] was selected because it possesses several salient features: (1) It has 14-dimensional geometry inputs, as shown in Fig 2, which is greater than other AEM studies found in the literature. The higher dimensional input grants more complexity. (2) The scattering response in this dataset is the absorptivity spectrum with 2001 frequency points and many sharp peaks that are traditionally challenging to fit. (3) This is the only dataset that is generated from full-wave simulation software. Each supercell, as shown in Fig 2, consists of four $SiC$ elliptical resonators. The geometry parameters of one supercell are: height $h$ (identical for all resonators), periodicity $p$, x-axis and y-axis radii $r_x, r_y$, and each elliptical resonator is free to rotate and described by $\theta$.

**Nanophotonic Particle. (Particle)** This problem was originally described and published in [21]. The nanophotonic particle is one of the first AEM studies used to learn the mapping $s = f(g)$ [21]. There are two favorable properties that the nanophotonics particle dataset possesses: (1) The geometry of the nanophotonics particle, shown in Fig 2, is a sphere consisting of alternating layers of $TiO_2$ and silica, which offers high flexibility for adjustment of the geometry input dimension. We set the

number of layers to eight in our benchmark in order to achieve a sufficient challenging problem for the surrogate model. (2) Output of the nanophotonic particle benchmark is the wavelength-dependent scatter cross-section, thus providing a slight variation of $s$. The large differences between ADMs and nanophotonic output can be used to validate the surrogate models' universality for AEM problems. In Fig 2 we show a four-layer particle for simplicity, but as mentioned, we explore the eight-layer structure, with each layer given by a thickness $r_1 - r_8$. The geometry parameters for the nanophotonic particle dataset thus consists of thickness values $r_1 - r_8$ for eight alternating $TiO_2$ and silica layers.

**Color filter. (Color)** The color filter problem was originally described and published in [40]. The color filter problem is interesting due to its unique color space output instead of transmission spectra. The color filter geometry is a three-layer Fabry–Pérot cavity, consisting of a $SiO_2$ layer sandwiched by two $Ag$ layers. When white light passes through the color filter, the three layers filter out wavelengths outside of the range of the resonance frequency – determined by the geometry $g$. The resulting transmission spectra yields a peak at the resonant frequency. The $SiO_2$ thickness primarily determines the resonate peak frequency, and the $Ag$ thickness fine tunes the full width half maximum of the resonant peak, thereby determining the corresponding color observed. The paper adopted the multiple beam interference formula to calculate transmission spectra [40]. The three geometry parameters for the color filter dataset are the thickness $t_1 - t_3$ for the bottom $Ag$ layer, $SiO_2$ layer, and top $Ag$ layer respectively.

## 4.2 Scoring Metrics

An important benchmark consideration is the chosen evaluation metric. Here we select the *average* mean squared error (MSE), given by,

$$e = \frac{1}{D_{out}} \frac{1}{N} \sum_{j=1}^{D_{out}} \sum_{i=1}^{N} (s_{ij} - \hat{s}_{ij})^2 = \frac{1}{D_{out}} \sum_{j=1}^{D_{out}} MSE(s_j, \hat{s}_j) \tag{5}$$

where $\hat{s}$ is the estimated AEM property, $i$ indexes the samples in the testing dataset, and $j$ indexes the dimensions of the output vector $s$ (often a spectrum). We select MSE because it is the most widely-used metric in the relevant AEM literature (e.g., [41]). It is also well-behaved and well-defined for all values of $s$, unlike the mean-relative-error – another metric sometimes used in the AEM literature (e.g., [21, 42]). MRE has the limitation that it grows exponentially as $s \to 0$, and becomes infinity when $s = 0$.

In addition to MSE, we have also included a wide range of metrics in our benchmark suite code: Mean absolute error (MAE), Mean absolute relative error (MARE), Mean squared relative error(MSRE), $R^2$, Kendall's Tau, Spearman's Rho. We have also put the result for Kendall's Tau and Spearman's Rho into the supplement for those who are interested in the ranking ability of these benchmarked algorithms.

## 4.3 Data Generation and Handling

The data associated with each benchmark problem was obtained by randomly and uniformly sampling geometries over some pre-defined domain, and then simulating these geometries to obtain their properties. Random sampling avoids any bias towards particular subsets of designs in the dataset, and is widely-used in the AEM literature. In each benchmark, we utilize the same sampling domain that was proposed in the original source publication. Further details about the sampling distributions can be found in Section 1 of the supplement.

To ensure a rigorous experimental process, we divided the total dataset available for each problem into three disjoint subsets: training, validation, and testing. Each of these three subsets are randomly sampled and disjoint. We first take out 10% of the data as the independent test set, to be used for model evaluation after training and hyperparameter optimizations are complete. We report the performance of optimized models on this independent test set. The remaining 90% of the dataset is divided into a training set (80%) and a validation set (20%).

## 4.4 Benchmark resources and extensibility

The code base for our benchmark task is maintained at the following remote repository: `https://github.com/ydeng-MLM/ML_MM_Benchmark`. The code base includes pipelines for the three

neural network architectures, and is open source under MIT license. Out of the three AEM benchmark problems, we generated the all-dielectric metasurface and nanophotonics particles datasets. They can be accessed [2] under the *CC0 1.0 Universal* license. We also cited the work from [40], and their color filter dataset can be accessed [3] under *CC by 4.0* license. All the datasets are hosted at a digital repository with permanent DOI under long-term preservation maintained by corresponding libraries. We plan to maintain our code base through a remote repository at github and pypi. Any issues regrading the software suite can be raised at the repository.

Although currently our benchmark suite only holds three datasets specifically from the AEM community and three neural network architectures, the benchmark suite's infrastructure is built to be highly extensible to new models and/or new datasets. New datasets can be put into the 'customize' folder and, following instructions, it is straightforward to conduct the same experiment on new dataset. Similarly, new models are easily extensible in our benchmark suite as clearly defined API and instructions are provided.

## 5    Numerical Experiments on the AEM Benchmark

In this section we utilize our proposed AEM benchmark to compare different approaches for data-driven surrogate modeling of AEM properties. For this purpose we employ a two-stage experimental design. In the first stage we aim to demonstrate the value of our benchmark by comparing a new model with an existing model, and also thereby establishing a strong baseline performance for each problem, upon which future work can build. To do this, we develop new surrogate models for each benchmark problem, and then compare the performance of our new models to that of the models originally reported for each problem - termed the "Baseline" models. The full details of each Baseline model (e.g., architecture and hyperparameter settings) can be found in Section 4 of the supplement.

Nearly all recent studies of data-driven surrogate models for AEM problems employ (deep) multi-layer perceptron (MLP) models, with the main differences arising in hyperparameter choices (e.g., model depth and width, learning rates), and the inclusion/exclusion of popular auxiliary processing strategies (e.g., dropout, batch normalization, skip connections). Rather than considering all of these variations as separate benchmark models, we begin by adopting the Baseline model originally proposed for each problem, and then performing a greedy step-wise optimization [43], whereby we optimize each of the model's hyperparameters, or consider including/optimizing auxiliary processing strategies,

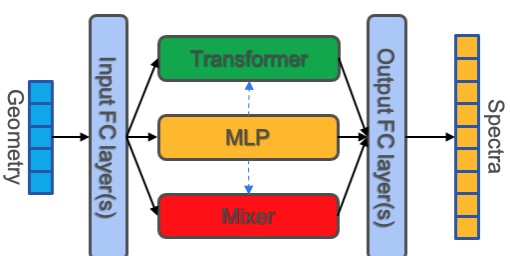

Figure 3: Schematic diagram of our adaptation of Mixer and Transformer to our data set. Here we use the optimized MLP model architecture as baseline to tweak our Mixer and Transformer models.

in turn. For this optimization we use a subset of the available training data as a validation dataset. Through this optimization we effectively benchmark a large number of existing models from the literature, and choose the best performer as the representative entry in our benchmark. We then compare the performance of our optimized model to the Baseline model, providing (to our knowledge) the first replication, and comparison with, an existing surrogate model and dataset within the AEM literature.

In the second stage of our experiments, we aim to explore the effectiveness of two recent DNN architectures as alternatives to the MLP: the Transformer[9] and MLP-Mixer[8]. Transformers are an attention-based deep learning architecture that has recently achieved state-of-the-art performance on a variety natural language processing (NLP) and computer vision tasks. The MLP-Mixer has recently achieved promising results as a simpler and more computationally-efficient alternative to the convolutional neural network [8]. We hypothesize therefore that the MLP-Mixer or the Transformer may also provide benefits for solving scientific computing problems, including AEM problems. By testing this hypothesis we also hope to contribute to the general machine learning community by considering whether the efficacy of these models extends to problems in scientific computing.

---

[2]https://doi.org/10.7924/r4jm2bv29
[3]http://dx.doi.org/10.5258/SOTON/D1686

To our knowledge, we are the first to consider Transformers (or any attention-based model) or MLP-Mixer models for surrogate modeling of AEM problems. In similar fashion to stage one of our experimentation, we also perform a greedy step-wise optimization of our Transformer and MLP-Mixer models for each benchmark problem, and subsequently compare their performance to the Baseline and optimized MLP models. Finally, we also evaluated several non-DNN-based models as additional baselines in our benchmark: Linear Regression, Random Forests and a Linear Support Vector Regression. Non-DNN-based models are rarely employed in the AEM literature as surrogate simulators, and the benchmark provides a useful opportunity to examine the validity of this practice.

## 5.1 Optimization of Models

In order to minimize potential bias across the three model optimizations that we performed (one for MLP, MLP-Mixer, and Transformer), we adopted two strategies. First, once we optimized the MLP model in stage one, we imposed that the Transformer and MLP-Mixer would not be allowed to exceed this size. This prevents the Transformer or MLP-Mixer from achieving better performance simply due to greater model capacity. We note however that we did allow for the MLP-Mixer or Transformer to have fewer parameters, if it was found that better performance was achieved. As a second strategy, we allotted a budget of 96 hours of optimization time (all on Nvidia 3090 GPUs), so that any one model would not benefit substantially from additional optimization effort.

## 5.2 Adaption of MLP-Mixer and Transformer AEM Problems.

Both the MLP-Mixer and the Transformer were originally designed for different tasks than ours, where the input data is structured (e.g., into sequences) and generally much higher in its dimensionality. To make our AEM problem more suitable, we leverage the fact that each of these network architectures is composed of layers that can (within some limits) be composed with one another (e.g., one can compose a Transformer and an MLP layer). Therefore, we adopt a strategy whereby we first process the input data, $g$, with several MLP (i.e., fully-connected layers) to extract a higher-dimensional representation of the input. Then we arbitrarily structure these new features into a sequence, so that it can be fed into Transformer and/or MLP-Mixer layers. This hybrid model architecture is outlined in Fig. 3. We provide the mathematical definition for each type of layer in Section 5.3, details of this proposed architecture can be found in the supplementary materials.

## 5.3 Deep neural network architectures

In this section we provide a mathematical definition of the three types of DNN layers that we employ in our benchmark experiments.

**MLP.** An MLP layer consists of a full matrix of connections between two sets of neurons. A conventional MLP layer with (optional) skip connections is given by

$$o_l = ReLU(W_l o_{l-1}) + o_{l-1} \tag{6}$$

where $l$ indexes the layer of the neural network and $o_l$ denotes the activations (or features) of layer $l$. $W_l$ refers to the weight matrix of layer $l$.

**Transformer[9].** The transformer layers employed in our experiments are defined as

$$u_i^{'} = \sum_{h=1}^{H} W_{c,h}^T \sum_{j=1}^{n} softmax\left(\frac{<W_{h,q}^T x_i, W_{h,k}^T x_j>}{\sqrt{k}}\right) W_{h,v}^T x_j, \tag{7}$$

$$z_i = LN(LN(x_i + u_i^{'}) + W_2^T ReLU(W_1^T LN(x_i + u_i^{'}))) \tag{8}$$

where $x, z \in \mathcal{R}^{n \times d}$ are input/output, $W_{h,q}, W_{h,k}, W_{h,v} \in \mathcal{R}^{d \times k}$ are weight matrices for query, key and value, $W_1 \in \mathcal{R}^{d \times m}, W_2 \in \mathcal{R}^{m \times d}$ are weight matrices for ending MLP inside encoder, LN is layer normalization.

**MLP-Mixer[8].** An MLP-mixer layer is defined as follows:

$$U_{*,i} = X_{*,i} + W_2 \times GeLU(W_1 LN(X)_{*,i}), \text{ for } i = 1 \text{ to } C, \tag{9}$$

$$Y_{j,*} = U_{j,*} + W_4 \times GeLU(W_3 LN(U)_{j,*}), \text{ for } j = 1 \text{ to } S, \tag{10}$$

where $W_i$ are weight matrices for the MLP and C and S are respectively hidden widths and number of patches that can be tuned, LN is layer normalization.

## 5.4   Results and Discussion

The results of our numerical benchmark experiments are presented in Fig 4, and a tabulation of the numerical values is provided in Table 3 of the supplement. First we note that the MLP model achieves similar or lower MSE than the Baseline model in all three benchmark problems. Furthermore, on the All-dielectric metasurfaces (ADM) and Particle problems, the MLP achieves a substantial reduction in MSE compared to the Baseline. This suggests that by systematically evaluating the efficacy on our benchmark of different modeling approaches from the literature, we were able to achieve consistent performance advantages.

The non-DNN-based models perform much worse than their DNN-based counterparts shown in Fig 4, and therefore we put the full details of their results in Fig. 3 of the supplement. DNN-based approaches now dominate as surrogate simulators in the AEM research and these benchmarking results provide some support for this practice, although further analysis with more non-DNN-based models would be beneficial.

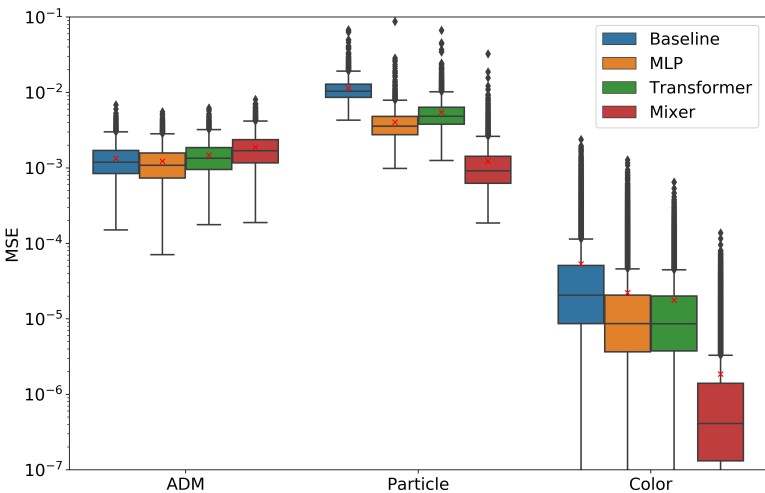

Figure 4: The box plot for MSE comparison between architectures. Baseline represents the original architecture shown in the paper that published the datasets. Mean values are reported as the red crosses. The image is slightly cropped as the lower percentile MSE values of the Color dataset being extremely low (a full version can be found in supplement Section 7 Figure 3).

The advantages of the MLP-Mixer and the Transformer, compared to the MLP, are less conclusive. The MLP-Mixer outperforms the MLP on the Particle and Color datasets, however it performs worst among all models on the ADM problem, albeit by a relatively narrow margin (based upon the variance of the MSEs). In no case did the Transformer achieve the lowest error, and it often achieves average performance among the models. Collectively these results suggest that the Transformer and MLP-Mixer do not offer consistent performance advantages over the MLP, however, the MLP-Mixer can sometimes yield substantial performance improvements depending upon the problem. We constructed a critical difference plot to quantitatively summarize the performance of each of these three architectures across the three benchmark problems, which is available in Fig. 4 of the supplement. This analysis suggests that there is no clear winner among the three architectures, corroborating our qualitative analysis.

Further insights into the results can be obtained by considering the model size (i.e., number of parameters) of each model in the benchmark, shown in Fig 5. Notably, the Baseline model is considerably smaller than the MLP, suggesting that previous models were too small for the complexity of the AEM problems. Furthermore, these results suggest that the performance advantage of the MLP relative to the Baseline (also an MLP) is likely driven in part by its greater size. By contrast, one clear advantage of the Transformer, and especially the MLP-Mixer, is its parameter efficiency. The MLP-Mixer requires substantially fewer parameters (and therefore computational efficiency) while offering similar or better performance than the other models.

## 6 Conclusions

In this work, we developed the first publicly available and easily accessible benchmark for data-driven surrogate modeling for AEM problems. Our benchmark includes three AEM problems that were adopted from existing work, and chose to be maximally relevant and representative of recent AEM research. We then used our benchmark to optimize and compare three different state-of-the-art deep learning architectures: the multilayer perceptron (MLP), MLP-Mixer, and Transformer. We also compare our developed models to existing models that were previously developed for each of our benchmark problems. To our knowledge we are the first to perform a systematic comparison of this time for AEM surrogate modeling, or to explore the use of MLP-Mixer and Transformer architectures for this problem.

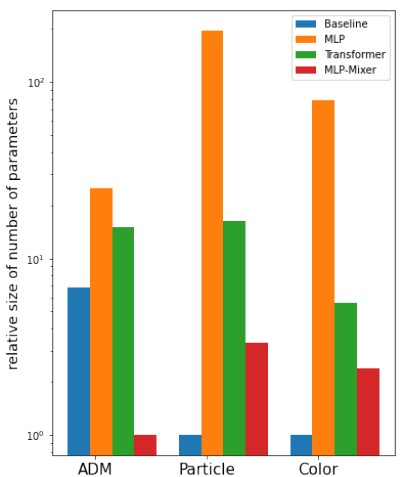

Figure 5: Relative number of parameters of optimized MLP, Transformer and MLP-Mixer models on each dataset.

## Acknowledgments and Disclosure of Funding

Yang Deng, Omar Khatib, and Willie Padilla would like to thank the funding from U.S. Department of Energy (DESC0014372).

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
