# OpenReview forum: "Benchmarking Data-driven Surrogate Simulators for Artificial Electromagnetic Materials"
_NeurIPS.cc/2021/Track/Datasets_and_Benchmarks/Round2 — NeurIPS 2021 Datasets and Benchmarks Track (Round 2)_

### Official Review · Reviewer_PUPL · 2021-09-13
**Interesting application but several aspects of paper lacking**

**Rating:** 4
**Confidence:** 3

**Strengths:**

* The paper helps bring materials design problems to the attention of the broader ML community in an accessible manner.
* Authors have significant domain knowledge and have provided thorough reasoning for the datasets they chose to develop.

**Weaknesses:**

* Since this is a ML conference, it should make sense for the tasks to be broadly presented to the ML community. This can happen if the problem structure is exposed in a way that supports interesting research directions in ML. For example, the presentation of molecules as graph-structured data spurred advances in GNNs and end-to-end learning from 3D structures. However, this paper frames the problem of material property prediction as a mapping from $\mathbb{R}^n \rightarrow \mathbb{R}^m$, with no additional structure, and so (from an ML perspective) is indistinguishable from the countless other application areas that can be so framed. For the same reason that using various classifiers (SVMs, MLPs, ...) with fixed-length molecular fingerprints is more a cheminformatics problem than a ML problem, it's not clear that the formal treatment of materials design in this paper is unifying and/or abstract enough to be an appropriate fit for NeurIPS.
    - The significance of the paper would be much improved if, for example, the authors leveraged the geometric nature of the input $g$, or the continuous stochastic process nature of the output function $s(\omega)$, or discussed ways of learning representations for a broad class of problems in AEM design.

* The choice of architectures to benchmark is unconvincing. The forced application of Transformers and MLP-mixers to partitions of a fixed-length input vector does not seem well-justified (as the authors themselves note). It would be much more interesting to compare with architectures that take 3D geometry into account, such as 3D CNNs, GNNs, point-cloud networks, etc...  Using such 3D methods seems particularly natural since CEMS involves solving Maxwell's equations in 3D.

* To summarize, because of the way the tasks are presented, it's not clear how pursuing further improvement on the datasets would be a fruitful line of ML methods development.

**Additional Feedback:**

The paper is not properly formatted. Authors used the preprint option instead of the submission option, and thus there are no line numbers.

**Clarity:**

Some aspects of the paper could be improved in clarity.
* How are the AEM designs in the dataset generated? Are they drawn from a distribution over the design parameters? What type of distribution?
* Authors should much more prominently (at the top of the corresponding sections) state when the datasets are original, when they are sourced from the previous work of the authors, and when they are sourced from other authors.
* The results in Figure 4 should be presented in tabular format for ease of comparison rather than only in a figure.
* What material are the layers of the nanophotonic particles manufactured from? Is this an input into the model?
* While I was generally able to follow how the three types of designs were featurized, the exposition could be clearer.

**Correctness:**

* How are the datasets split into train/val/test? This is a critical process---which affects the correctness of the submission---that should be described and justified in detail.
* Minor correction: [MLP-mixer](https://arxiv.org/abs/2105.01601) is proposed as an alternative to CNNs, not transformers
* The "performance bias mitigation" seems to be an unconventional restriction that could affect the correctness of results. I am particularly concerned about the asymmetry of the restriction. It doesn't look like it affected any results, but I would suggest standardizing for model size in a more systematic and symmetric manner.

**Documentation:**

As discussed above, a number of details on data collection/generation are missing.

The datasets are provided in an accessible manner. Documentation is sufficient but not outstanding.

Benchmarks appear easily reproducible.

**Ethics:**

No ethical concerns.

**Relation To Prior Work:**

No concerns.

**Summary And Contributions:**

The authors present datasets for 3 tasks in the design of Artificial Electromagnetic Materials (AEMs)---materials with useful electromagnetic properties based on their nanoscale geometry---and benchmark 3 common deep learning architectures on these tasks. While there has been much interest and progress in using deep learning, or "surrogate models", to approximate these structure-property relationships, there have been almost no datasets made available, making replication of results very difficult. Authors propose to establish a groundwork for this emerging application area by releasing datasets for 3 relevant AEM design problems. Authors benchmark MLPs, MLP-mixers, and transformers on these tasks, with significant modification to the latter two architectures to accept fixed-length feature vectors describing the material geometry.

---

> ### Author Response · Authors · 2021-09-30
> **Response to reviewer PUPL part1**
>
> We thank the reviewer for taking the time to review our paper!  Below we list each of the reviewer’s comments (italicized) and then provide our response (not italicized).  Some reviewer comments are related, and therefore we provide one response after listing two consecutive reviewer comments.
>
>
> >1. *Since this is a ML conference, it should make sense for the tasks to be broadly presented to the ML community. This can happen if the problem structure is exposed in a way that supports interesting research directions in ML. For example, the presentation of molecules as graph-structured data spurred advances in GNNs and end-to-end learning from 3D structures. However, this paper frames the problem of material property prediction as a mapping from R^n->R^m, with no additional structure, and so (from an ML perspective) is indistinguishable from the countless other application areas that can be so framed. For the same reason that using various classifiers (SVMs, MLPs, ...) with fixed-length molecular fingerprints is more a cheminformatics problem than an ML problem, it's not clear that the formal treatment of materials design in this paper is unifying and/or abstract enough to be an appropriate fit for NeurIPS.*
>
> >2. *To summarize, because of the way the tasks are presented, it's not clear how pursuing further improvement on the datasets would be a fruitful line of ML methods development.*
>
> Response:  Although there are many ML problems of the form R^{n} -> R^{m}, we must respectfully disagree that this makes all such problems “indistinguishable”.  Note that this same criticism can be applied to the 3D formulation proposed by the reviewer - there are, by now, a large number of ML problems of this input/output form, but this does not make them all indistinguishable.  Even when problems have the same input/output formulation, they are still different from one another because (at a minimum) the underlying functions that one is trying to approximate/infer are different (e.g., in medicine, remote sensing, materials science), necessitating the development of unique ML models.  This is one important reason (among others) that these problems are not indistinguishable from one another, and why our benchmark is not indistinguishable from other problems of the form R^{n} -> R^{m}.
>
> For the same reasons above, there is still substantial room for useful experimentation and innovation for our problem, despite its R^{n} -> R^{m} formulation.   As noted by the reviewer, there are innumerable problems of this form, and research continues despite the fact that these problems all share the same input/output formulation ( R^{n} -> R^{m} ), suggesting that there is still substantial room for innovation.  In particular, in artificial electromagnetic material problems we are trying to build ML models that approximate solutions to Maxwell’s equations, requiring unique modeling solutions.
>
> It may be the case that an alternative formulation of the problem (e.g., with 3D input, as the reviewer has suggested) would be a valuable research direction.  However, this approach has drawbacks, particularly for benchmarking purposes.  For example, the vast majority of existing AEM research utilizes the formulation in our benchmark, making it more relevant and attractive to researchers working at the intersection of ML and electromagnetic material design - this was a major motivation for using our current formulation, and it represents a major advantage over alternatives.  We note too that it is still possible for researchers to explore the use of alternative formulation (e.g., 3D input) on our benchmark - sufficient information about each benchmark problem is provided in the paper to support exploration of these ideas.  Furthermore, the existence of our benchmark will make it easier to directly compare and demonstrate the effectiveness of any alternative formulations.

---

> ### Author Response · Authors · 2021-09-30
> **Response to reviewer PUPL part2**
>
> >3. *The choice of architectures to benchmark is unconvincing. The forced application of Transformers and MLP-mixers to partitions of a fixed-length input vector does not seem well-justified (as the authors themselves note). It would be much more interesting to compare with architectures that take 3D geometry into account, such as 3D CNNs, GNNs, point-cloud networks, etc... Using such 3D methods seems particularly natural since CEMS involves solving Maxwell's equations in 3D.*
>
> Response:  We agree that some modification of transformers and MLP-mixers was needed to apply them to our problem, however, we don’t think that this implies that using these models is unjustified, or arbitrary.  In Section 5, paragraph 3, of the manuscript we explain the hypotheses that motivated our use of Transformers and MLP-Mixers, providing justification for their selection (i.e., they are not purely arbitrary choices).  Furthermore, the adaptation of models to work on new problems is a common feature of ML research, and represents one form of innovation.  For example, the recent and well-cited TransUNet model (Chen et al., 2021) applies transformers to medical image segmentation, and utilizes a similar strategy to the one we use here; they apply transformers to the high-dimensional features produced after several conventional neural network layers.
>
> >4. *How are the datasets split into train/val/test? This is a critical process---which affects the correctness of the submission---that should be described and justified in detail.*
>
> Response:  We agree that these are important details and we thank the reviewer for pointing this out.  We have added this information into the revised manuscript in section 4.3.   In summary, we allocate a random 10% of the total data into a test set, and then we split the remaining data into a training dataset (80%) and a validation dataset (20%), for optimizing the models.
>
> >5. *Minor correction: MLP-mixer is proposed as an alternative to CNNs, not transformers*
>
> Response:  Thank you for pointing this out. We have corrected our description in section 5, paragraph 3, of the manuscript.
>
> >6. *How are the AEM designs in the dataset generated? Are they drawn from a distribution over the design parameters? What type of distribution?*
>
> Response:  Based upon the reviewer’s comment, we realize this important information is not present, and we have now added a section in the main manuscript entitled “Data Generation and Handling” (Section 4.3 in the revised manuscript) where we address these questions.
>
> >7. *Authors should much more prominently (at the top of the corresponding sections) state when the datasets are original, when they are sourced from the previous work of the authors, and when they are sourced from other authors.*
>
> Response:  We agree, and we have added clarifications about the origins of our benchmark problems in section 4.1 at the beginning of paragraph 3, 4, and 5, respectively.
>
> >8. *The results in Figure 4 should be presented in tabular format for ease of comparison rather than only in a figure.*
>
> Response:  Thank you for the suggestion - we have put the tabular format of the result comparison in supplement section 5 as table 3 for easy reference. We also mentioned these supplementary results for readers in the main manuscript section 5.4, paragraph 1.
>
> >9. *What material are the layers of the nanophotonic particles manufactured from? Is this an input into the model?*
>
> Response: The nanophotonic particles have a silica core and alternating layers of TiO2 and silica. The description is added to the manuscript in section 4.1,  paragraph 4.  Although the material composition could conceivably be added to the model, this was not included in the original formulation of this problem by its original authors, and therefore we did not include it as input to the models in our benchmark.
>
> >10. *While I was generally able to follow how the three types of designs were featurized, the exposition could be clearer.*
>
> Response:  Thank you for bringing this to our attention.  We agree that this information was either unclear or incomplete in some cases.   Based upon the reviewer’s feedback we have revised our description of each dataset in section 4.1 to clarify the composition of the features that were input to each model.   We have also added further details about each dataset to Section 1 of the supplement.

---

> ### Comment · Reviewer_PUPL · 2021-10-04
> **Reviewer response**
>
> I thank the authors for updating the manuscript and addressing many of the concerns raised.
>
> However, I remain unconvinced that the problem is formulated in a way that makes it suitable for a track at an ML conference. I will defer to the judgement of the area chairs, but my opinion is that the framing of problem as $\mathbb{R}^m \rightarrow \mathbb{R}^n$ (with hand engineered features) is insufficiently well-structured, interesting, or novel that it would serve as a motivator of interesting research in ML, when considering the immense body of work on problems of that type over many decades, and the immense number of problems that, with hand-engineered features, could be so framed. However, the underlying scientific question is very interesting and it would be fascinating to see ML adapted for end-to-end learning on that domain. I believe a submission approaching the problem from that angle would be more compelling.
>
> RE. authors' points: I understand that convention in the field may have been to formulate the problem in this way, but just as applying GNN methods to molecules was initially a novel idea, deviating from convention here does not necessarily mean less scientific impact. Additionally, current fruitful lines of ML work deal strongly with notions of problem structure---inductive biases over end-to-end models---which is not exposed by the current formulation. I agree that the formulation (and dataset) proposed by the authors has value, but without an articulation of the problem structure, it is more interesting to domain practitioners than to the ML research community.

---

### Official Review · Reviewer_FfhS · 2021-09-20
**DL for AEMs**

**Rating:** 8
**Confidence:** 4
**Clarity:** Yes the paper is well written.

**Strengths:**


- At the time of review, the repository provides a straightforward way to download the datasets. An implementation of the 3 models discussed are also available along with pre-trained weights.
- The choice of datasets appears well motivated and there is a brief discussion of each. The authors have included their selection criteria for the datasets which ensure both relevance and representativeness.  The authors also include significant detail on the data generating process as well as a brief overview of the theoretic background.
- A baseline model is included as representative of prior work, which provides a context for judging the performance of newer models.

**Weaknesses:**


- There is little to no information on how splitting was conducted for these benchmark datasets, which leaves the reader to assume that a random split was conducted. Information about split sizes and scattering distributions is also missing.
- Baseline model for each dataset deserves more discussion or at least a brief description of the hyperparameters. It is hard to place results in Figure 4 in context. It is unclear how different the Baseline model is compared to the MLP model for example.



**Additional Feedback:**

- Figure 4
    - the Color dataset results seem to be cropped slightly
- Figure 5:
    - Not sure which best performance means in this case? Would that mean lowest average MSE across datasets? That would just mean that number of parameters are log ratio'd by their performance on the Color dataset, which seems like a weak comparison.
- Supp. Figure 2:
    - missing a-f labels.

Minor Comments

- 3.3: "a parameterized geometry g – which is of order g ∼ 101 – of AEMs as input"
    - slightly confusing. would suggest either using some notation to indicate that the dimensionality of g is on the order of 10s or remove the em dash entirely
- Missing grid search results. The authors state in Section 5 that they performed hyperparameter optimization to find the best performing model using a validation set. It would be helpful to ML practitioners as well as strengthen the paper to include these results.

**Correctness:**

The information presented appears to the best of the reviewer's understanding of the AEM field to be correct. Information about model description is sound.

**Documentation:**

The documentation is good.

**Relation To Prior Work:**

There are no existing benchmark datasets for ML in this field.

**Summary And Contributions:**

The paper is well motivated as the authors point to the lack of shared models and datasets despite the rise in number of papers which apply deep learning to this problem. Application of deep learning to this domain is an instance of avoiding high compute by instead training a surrogate deep learning model on a set of observations. The resulting speedup benefits the design of artificial electromagnetic materials. However without standardized benchmarks, there is no clear route to assessing the generalizability of models. This issue  also hinders reproducibility and limits the impact of improvements in the field. This work aims to introduce a 3 benchmark datasets from within the field of AEMs as well as provide some initial evaluation of model performance.

---

> ### Author Response · Authors · 2021-09-30
> **Response to reviewer FfhS**
>
> We thank the reviewer for taking the time to review our paper! Below we list each of the reviewer’s comments (italicized) and then provide our response (not italicized).
>
> >1. *There is little to no information on how splitting was conducted for these benchmark datasets, which leaves the reader to assume that a random split was conducted. Information about split sizes and scattering distributions is also missing.*
>
> Response:  We agree that these are important details and we thank the reviewer for pointing this out.  We have added this information into the revised manuscript in section 4.3.   In summary, we allocate a random 10% of the total data into a test set, and then we split the remaining data into a training dataset (80%) and a validation dataset (20%), for optimizing the models.
>
> >2. *Baseline model for each dataset deserves more discussion or at least a brief description of the hyperparameters. It is hard to place results in Figure 4 in context. It is unclear how different the Baseline model is compared to the MLP model for example.*
>
> Response:  We agree, and based upon the reviewer’s feedback, we have created a new section in the Supplement (Section 4), providing the following information: The baseline model architectures, hyperparameters for each baseline models, and the reference back to the original paper that we based on to test the baseline model.  We also added a sentence in Section 5.4, Paragraph 1, referring readers to this information in the supplement.
>
> >3. *Figure 4: the Color dataset results seem to be cropped slightly/Figure 5: Not sure which best performance means in this case? Would that mean lowest average MSE across datasets? That would just mean that number of parameters are log ratio'd by their performance on the Color dataset, which seems like a weak comparison./Supp Figure 2: missing labels*
>
> Response:  Thank you for bringing these figure problems to our attention. We have revised figure 4 and 5 captions for clarity in the main manuscript and added the label for supplement figure 2.  For figure 5, it is the relative number of parameters (in log scale) within each benchmark dataset (for the same dataset, the relative ratio between the number of trainable parameters of each architecture).

---

### Official Review · Reviewer_uskM · 2021-09-20
**Well thought-through collection of datasets and valuable benchmarks**

**Rating:** 8
**Confidence:** 2

**Strengths:**

- The paper makes a strong case for the value of the benchmark by pointing out the gaps in the field.
- The study is rigorous and appears well thought-through from a general ML perspective (my knowledge of metamaterials is not sufficient to judge from the domain perspective).
- The software allows for future additions to the benchmark datasets. This is important as the field progresses.

**Weaknesses:**

The benchmarks do not contain a simple non-deep learning baseline. It would be great if the authors could add one or comment on why only DL models are suitable for the presented tasks.

**Additional Feedback:**

-

**Clarity:**

The paper is very clearly written and explains the problems well to non-domain experts.

**Correctness:**

Experiment design is appropriate and has been performed according to well-established standards. Datasets were adopted from previous publications.

**Documentation:**

Software, datasets and pre-trained models are readily available.


**Ethics:**

No ethical concerns.

**Relation To Prior Work:**

The paper discusses an ample body of previous work and justifies its claims. I am not familiar enough with metamaterials though to judge whether something important is missing.

**Summary And Contributions:**

This paper presents three benchmarks for artificial electromagnetic materials: one each on metamaterials, nanophotonics, and color filter designs. The datasets and code are openly available.

Edit: I have updated the score after revisions.

---

> ### Author Response · Authors · 2021-09-30
> **Response to reviewer uskM**
>
> We thank the reviewer for taking the time to review our paper! Below we list each of the reviewer’s comments (italicized) and then provide our response (not italicized).
>
> >1. *The benchmarks do not contain a simple non-deep learning baseline. It would be great if the authors could add one or comment on why only DL models are suitable for the presented tasks.*
>
> Response:  The vast majority of AEM literature employs DL models for surrogate simulation, and that was the primary motivation for our focus on DL models (as a start) for our benchmark.  For example, the three benchmark problems that we included were obtained from recent publications from different authors, and in each case these papers all employed DL models.  However, it is plausible that DL models are not always the best choices, and therefore we would agree that there is great value in examining this hypothesis by including some non-DL models in the benchmark.  Based upon the reviewer’s feedback, we have examined the performance of three non-DL models (Linear Regression, (Linear) Support Vector Regressor and Random Forests).  Although other non-DL models may perform better than these, we found that none of these models outperformed any of the DL models in our benchmark, and we have added these results to appendix section 6.  We reference these results in the main text in section 5 paragraph 4 so that readers are aware of these results.

---

> > ### Comment · Reviewer_uskM · 2021-10-04
> > **follow up**
> >
> > Thanks for adding the baselines and for explaining. This improves the paper from a non-domain expert’s perspective.

---

### Official Review · Reviewer_uT1t · 2021-09-21
**towards making AEM benchmarking more accessible**

**Rating:** 6
**Confidence:** 2

**Strengths:**

* In the field of AEM research, papers that publish code and datasets are apparently rare. Relevance to this community appears to be given.


**Weaknesses:**

* The suite appears to be quite limited in the number of problem classes (3) and so does the experimental section (comparing baseline, MLP, Transformer and Mixer).
* It may be useful to further investigate other performance metrics than only the average MSE. If the models are indeed used as surrogate models as described in Section 3.3., other metric such as Spearman's Rho or Kendall's Tau may be of additional interest.
* The analysis of results is solely based on visually inspecting performance, as given in Figure 4 in the form of box plots of the MSE. It could be interesting to additionally perform statistical comparison, for example in the form of critical difference plots.


**Additional Feedback:**

* I do must admit that I have very little experience regarding the field of artificial electromagnetic materials (AEMs), therefore my review mostly focuses on the general benchmarking part of the paper.

**Clarity:**

* The paper is written well and can be followed along easily.
* Maybe some more focus could be given to the concept of using different DNNs as surrogate models of the AEM problem.

**Correctness:**

* Data appears to be constructed in a sound way.
* The benchmark design is reasonable, although only one performance metric is evaluated (MSE).

**Documentation:**

* Detail on data collection is given.
* Data is available on different hosting sites. Note that the first URL linking to the ADM dataset appears to be broken.
* A GitHub repository is also linked providing the code used for the benchmarks in this paper.
* Furthermore, the pre-trained models are available.
* Licenses are started for all data and code.
* No explicit maintenance plan is given.


**Ethics:**

* NA

**Relation To Prior Work:**

* Related work is discussed and focus is given to the lack of publicly available AEM problems.

**Summary And Contributions:**

A collection of three classes of AEM problems is released and accompanied by a GitHub repository.
Two problems are completely novel, whereas the third one uses existing resources.
AEM problems were selected based on relevance, representativeness and span of complexity.
Three different DNNs are then exemplarily benchmarked as potential surrogate models on each of the AEM problems.

---

> ### Author Response · Authors · 2021-09-30
> **Response to reviewer uT1t**
>
> We thank the reviewer for taking the time to review our paper! Below we list each of the reviewer’s comments (italicized) and then provide our response (not italicized).
>
> >1. *The suite appears to be quite limited in the number of problem classes (3) and so does the experimental section (comparing baseline, MLP, Transformer and Mixer).*
>
> Response:  Regarding our selection of models, the vast majority of models in the AEM literature are conventional deep learning models (especially deep MLPs), and most of these models are very similar to one another.  Therefore, to approximate a benchmark comparison of the models in the literature we performed a stepwise optimization of our MLP models on each dataset separately, sweeping over many of the parameters that vary among models in the literature.  Therefore the effective number of models from the literature compared in the benchmark is much larger than it may appear from our table of results (this rationale is described in Section 5, paragraph 2).
>
> This being said, we agree with the reviewer that more models (and datasets) would increase the value of our benchmark.  In response to the reviewer’s feedback, we have evaluated three additional non-deep-learning models on our benchmark problems (Linear Regression, (Linear) Support Vector Regressor, and Random Forests).  Although other non-DL models may perform better, we found that none of these models outperformed any of the deep learning models in our benchmark.  Therefore, we have added these new results to appendix section 6, however, we reference them in the main text in section 5 paragraph 4 so that readers are aware of them.
>
> It is unfortunately much more difficult to add additional datasets due to the time/expertise required to prepare and generate simulations; as well as the time required to benchmark models (especially deep learning models) on any new datasets.  Therefore we could not enhance our benchmark with more datasets within the review period.  While we agree that more datasets would enhance the benchmark, we still believe our benchmark represents a significant contribution since it is the first of its kind in this literature (which is a rapidly-growing community, as shown in Fig. 1), and we carefully selected the three benchmark datasets to be relevant, diverse, and representative of contemporary problems of interest (see descriptions in Section 4.1).
>
> >2. *It may be useful to further investigate other performance metrics than only average MSE. Other metrics such as Spearman’s Rho or Kendall’s Tau may be of additional interest*
>
> Response: Thank you for your suggestion. We agree that extra metrics would provide much richer information regarding the performance of the models, and aid comparison.  In response to the reviewer’s feedback, we have added Spearman’s Rho, Kendall’s Tau to the appendix section 7 and mentioned it in the main text section 4.2 paragraph 2.  In summary, and fortunately, the results from these two metrics are highly correlated with our MSE results.
>
> >3. *The analysis of results is solely based on visually inspecting performance, as given in Figure 4 in the form of box plots of the MSE. It could be interesting to additionally perform statistical comparison, for example in the form of critical difference plots.*
>
> Response: We agree with the reviewer and we think that a more quantitative comparison would be helpful.  In response to the reviewer we prepared histograms of the MSEs of the models, and the critical difference plots in the appendix, section 3 & 7 respectively.  In summary, the distances are relatively large and suggest that there is no clear winner, corroborating the qualitative analysis.  In the main manuscript in Section 5.4, Paragraph 2, we now refer the reader to these results, and explain that they corroborate our qualitative analysis.
>
> >4. *No explicit data/code maintenance plan is given*
>
> Response: Thank you for pointing this out - we agree.  Based upon the reviewer’s feedback we have updated the paper with a maintenance plan.  In particular, we have added a maintenance plan to the manuscript in section 4.4 paragraph 1.  For the datasets, two of them are hosted on Duke Research Data repository with a permanent DOI associated with it. It also has a contact person attached to it so any broken link or problem can be solved with contact. For the code, it is on github with pypi hosting and permits issues to be raised there.

---

### Decision · Program_Chairs · 2021-10-10

**Decision:**

Accept

**Comment:**

This paper proposes a dataset and a benchmark for materials design. It is an important contribution that would allow ML innovations in this important area of research. The reviewers mostly agree that the paper is well written, introducing material design well to the ML audience. The dataset is well documented and formatted, and appropriate baselines are presented. This paper is an important contribution to multi-disciplinary research in an important area of science, so I recommend acceptance.